# Seasonal Temperature Fluctuations Differently Affect the Immune and Biochemical Parameters of Diploid and Triploid *Oncorhynchus mykiss* Cage-Cultured in Temperate Latitudes

**Andreia C. M. Rodrigues** [1,*], **Carlos Gravato** [2], **Carlos J. M. Silva** [1], **Sílvia F. S. Pires** [1], **Ana P. L. Costa** [1], **Luís E. C. Conceição** [3], **Paulo Santos** [4], **Benjamín Costas** [4,5], **José Calheiros** [6], **Manuela Castro-Cunha** [6], **Amadeu M. V. M. Soares** [1] **and Rui J. M. Rocha** [1]

1   CESAM—Centro de Estudos do Ambiente e do Mar & Departamento de Biologia, Universidade de Aveiro, Campus Universitário de Santiago, 3810-193 Aveiro, Portugal; carlosjmsilva@ua.pt (C.J.M.S.); silviapires1@ua.pt (S.F.S.P.); anaplcosta@ua.pt (A.P.L.C.); asoares@ua.pt (A.M.V.M.S.); ruimirandarocha@ua.pt (R.J.M.R.)
2   Faculdade de Ciências da Universidade de Lisboa & CESAM, Universidade de Lisboa, Campo Grande, 1749-016 Lisboa, Portugal; cagravato@fc.ul.pt
3   Sparos, Lda. Área Empresarial de Marim, Lote C, 8700-221 Olhão, Portugal; LuisConceicao@sparos.pt
4   Centro Interdisciplinar de Investigação Marinha e Ambiental (CIIMAR/CIMAR), Universidade do Porto, 4450-208 Matosinhos, Portugal; paulo.santos@ciimar.up.pt (P.S.); bcostas@ciimar.up.pt (B.C.)
5   Instituto de Ciências Biomédicas Abel Salazar, Universidade do Porto, 4050-313 Porto, Portugal
6   A. Coelho & Castro Lda. Praça Luís de Camões, 4490-441 Póvoa de Varzim, Portugal; quintadosalmao@mail.telepac.pt (J.C.); acoelhocastro@mail.telepac.pt (M.C.-C.)
*   Correspondence: rodrigues.a@ua.pt

**Abstract:** In the coming decades, and despite advances in the selection of resistant strains and the production of triploid organisms, the temperature could seriously affect salmonid aquaculture. Lower environmental tolerance has been hinted at for triploids, but the physiological mechanisms leading to such differences, and whether they are translated to the individual level, are poorly understood. This study aimed to evaluate the effects of seasonal variations on the humoral and immune status in the blood (peripheral blood leukocytes) and plasma (antiprotease, lysozyme and peroxidase activities), the oxidative stress (catalase, glutathione-S-transferase, total glutathione and lipid peroxidation) balance in the liver, and the energy budget (sugars, lipids, proteins and energy production) in the liver and muscle of diploid and triploid *Oncorhynchus mykiss*. Leukocytes' numbers changed with the water temperature and differed between fish ploidies. Peroxidase activity was increased in the summer, but lysozyme and antiprotease activities were increased in the winter. Concomitantly, antioxidant defenses were significantly altered seasonally, increasing oxidative damage at higher temperatures. Moreover, warmer waters induced a reduction in the energy production measured in the liver. Differences in feed efficiency, which have been previously reported, were confirmed by the low lipid and protein contents of the muscle of the triploids. In sum, the inherent trade-offs to deal with the seasonal changes culminated in the higher growth observed for diploid fish.

**Keywords:** rainbow trout ploidy; water temperature; immunology; oxidative stress responses; energy budget

---

## 1. Introduction

Aquaculture production continues to grow in order to satisfy the global human demand for food, since fish protein ensures an energy source of easy digestion and high quality. In 2016, fish production accounted for around 171 million tonnes of food, with the total first-sale value of the production estimated at USD 232 billion, representing 47% of fish production [1]. Among farmed fish, *Oncorhynchus mykiss* production accounted for ~814,000 tonnes, corresponding to USD 3409 million in 2016 [2]. Despite being a cold-water species, an adaptation to warm water has been observed in rainbow trout farmed in the northern hemisphere, with optimum metabolic performances described at ~15 °C [3]. However, in the actual scenario of climate change, dramatic fluctuations in water parameters are taking place, such as higher peak temperatures during the summer period [4].

*O. mykiss* triploid fish started to be created in the middle of the 1980s, by thermal or pressure shocks applied to eggs shortly after fertilisation [5]. However, their lower growth and survival in warmer waters were quickly noted [6]. There is also rising evidence of the increased costs of triploidy, due to triploids' higher mortality rates and underperformance in suboptimal conditions, namely suboptimal temperatures, when compared to diploids [7–11]. The better survival rates of triploid trout were only observed in fish reared for recreational purposes at low densities [12]. However, the biological mechanisms instigating this weaker performance are still unclear, and may be related to the different cell size of the triploid fish. Triploids have increased nuclear and cellular volume in order to accommodate 50% more genetic material than diploids, which often implies a reduction in total cell numbers (review by [7]). Induced triploidy usually leads to a sterile fish, which is an advantage in aquaculture, since sterility ensures somatic growth and flesh quality, instead of the undesirable effects of energy allocation for sexual maturation, and prevents the establishment of escapees outside their native range [7,12]. Despite the reported potential advantages having lead to an amplified use of triploids in aquaculture, the study of triploid cells' physiology is still in its infancy [13]. Thus, attaining a better understanding of the fish's immune system and physiology is of utmost importance for adequate welfare and health management in sustainable aquaculture [14,15].

Therefore, this study aimed to contribute to a broader insight into physiological and metabolic differences between diploid and triploid *O. mykiss* reared under intensive culture conditions and facing real seasonal fluctuations. The animals were monitored over a year, and six sampling periods—representative of the water temperature's fluctuation range—were selected. The fish's cellularity (as obtained via blood smears) was used as a quick and representative away to assess the immunological status of the fish. In order to identify periods of immunosuppression, the humoral parameters (lysozyme, peroxidase, and antiprotease activities) were measured in each group. Biochemical responses related to oxidative stress status reflect the equilibrium between reactive oxygen species (ROS) production and the organism's capability to mitigate their damaging effects; the energy budget to maintain cellular metabolism and homeostasis were also investigated.

## 2. Materials and Methods

### 2.1. Fish Rearing Conditions and Sampling

*Oncorhynchus mykiss* (Walbaum, 1792) is a fusiform fish of the Actinopterygii class, the Teleostei infra-class, and the Salmonidae family. Two batches of diploid and triploid *O. mykiss* juveniles were reared and sampled at Quinta do Salmão farm at Rabagão river dam (Montalegre, Portugal), from June 2016 to January 2017 (Figure 1). Fry individuals with an average weight of 7.3 ± 2.0 g and a total length of 7.9 ± 0.9 cm, obtained from different broodstock, were selected and acclimated to farm conditions for 3 weeks prior to the beginning of the monitoring period. The fish were reared in 16 m diameter circular cages, with an average density of 12–15 kg/m$^3$. The animals were fed twice a day, with 0.7–2.3% alpis extruded feed. Six representative points of the annual cycle temperature fluctuation were selected, and ten fish per batch were sampled in the morning period at each sampling time (Figure 1). The fish were sacrificed using an overdose of the anaesthetic 3-aminobenzoic acid ethyl

ester (MS-222, 0.1 g/L), weighed, measured, and their blood was quickly collected from the caudal vein using heparinised syringes. A piece of liver and muscle tissues were also removed from each fish and stored in liquid nitrogen for posterior analysis of their biochemical biomarkers. The experimental procedures were previously submitted for approval by the national competent authorities, and the manipulation of the animals was performed by qualified scientists following the national rules and the European Directive 2010/63/EU of the European Parliament and the European Union Council on the protection of animals used for scientific purposes.

The fish were cultured following the standard protocols adopted by the company, under a natural photoperiod. The dissolved oxygen values for the dam where the aquaculture is located vary between 5 mg/L in warmer months and 10 mg/L in colder periods (https://snirh.apambiente.pt/).

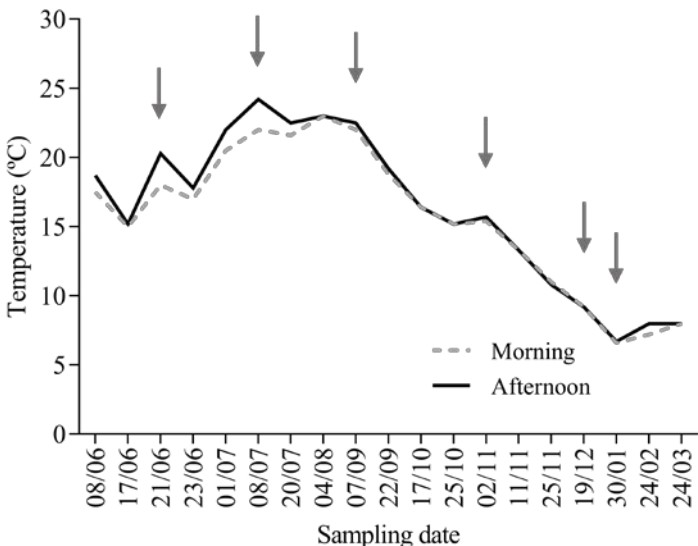

**Figure 1.** Fish sampling data and the temperature (°C) conditions of the water during the morning and afternoon, as measured in the cages located at the Rabagão river dam from June 2016 until March 2017. The arrows point to the temperature in the days of sampling.

## 2.2. Hematological Procedures

The blood smears were conducted according to Machado et al. [16]. The slides were air dried, and then fixed with a solution of formaldehyde-ethanol (90% absolute ethanol to 10% of 37% formaldehyde) for 1 min [17]. The neutrophils were identified by the detection of peroxidase activity, following Afonso et al. [18]. Later, the slides were stained with Wright's stain (Haemacolor, Merck) and observed under oil immersion (1000×). The immune cells were identified, and the neutrophils, monocytes, lymphocytes, and thrombocytes were differentially counted in a total of 200 cells/smear.

## 2.3. Innate Humoral Parameters

The antiprotease activity was determined as described by Ellis [19], adapted for 96-well microplates [16]. The plasma (10 μL) was incubated (10 min, 22 °C) after adding the same volume of trypsin solution (5 mg/mL in $NaHCO_3$, 5 mg/mL, pH 8.3) into polystyrene microtubes. To the incubation mixture, 100 μL phosphate buffer ($NaH_2PO_4$, 13.9 mg/mL, pH 7.0) and 125 μL azocasein (20 mg/mL in $NaHCO_3$, 5 mg/mL, pH 8.3) were added and incubated (1 h, 22 °C). According to the protocol described by T. Ellis, both trypsin and azocasein are dissolved in a 5 mg/mL $NaHCO_3$ solution. Since both solutions have the same concentration, once mixed, the $NaHCO_3$ concentration remains the same. After this, 250 μL trichloroacetic acid 10% (100 mg/mL) was added to the microtubes and incubated (30 min, 22 °C). Finally, the mixture was centrifuged at 10,000× g for 5 min. Phosphate buffer saline—instead of plasma and trypsin—was used for the blank, and phosphate-buffered saline

was used instead of plasma to obtain the reference sample. The percentage of trypsin activity was calculated using the following formulas:

$$\% \text{ non-inhibited trypsin} = (\text{Sample absorbance} \times 100)/\text{Reference sample}$$

$$\% \text{ inhibited trypsin} = 100 - \% \text{ non-inhibited trypsin}$$

The lysozyme activity was measured using a turbidimetric assay, as presented in Costas et al. [20]. A solution of *Micrococcus lysodeikticus* (0.5 mg/mL, 0.005 M sodium phosphate buffer, pH 6.2) was prepared. The plasma (15 μL) and the *M. lysodeikticus* suspension (250 μL) were added to a microplate in order to achieve a final volume of 265 μL. The reaction was carried out, and the absorbance (450 nm) was measured after 0.5 and 4.5 min at 25 °C. The standard curve of a lyophilised hen's egg-white lysozyme (L3790, Sigma) was used.

The total peroxidase activity in the plasma was measured following Quade and Roth [21]. Firstly, the samples were diluted in Hank's Balanced Salt Solution (HBSS) on a 1:10 dilution. The diluted samples (15 μL) were added to 250 μL of HBSS without $Ca^{2+}$ and $Mg^{2+}$ in flat bottomed 96-well plates. Then, 20 mM 3,3′,5,5′-tetramethylbenzidine hydrochloride (50 μL, TMB, Sigma) and 5 mM hydrogen peroxide (50 μL) was added to the mixture. The colour change reaction was stopped after 2 min by adding 50 μL 2 M sulphuric acid, and the optical density was read at 450 nm. The wells without plasma were the blanks. The peroxidase activity (units/mL plasma) was determined to define one unit of peroxidase as the amount of peroxidase that produces an absorbance change of 1 Optical Density (OD).

All of the readings were performed in a Synergy HT microplate reader, Biotek.

## 2.4. Sample Preparation for Oxidative Stress and Cellular Energy Allocation Analysis

The fish tissue (liver and muscle) samples were individually homogenised by sonication (pulsed mode of 10% for 30 s, 250 Sonifier, Branson Ultrasonics) on ice, using ultra-pure water (1600 μL). From the liver samples, one aliquot containing 4% butylated hydroxytoluene (BHT) in methanol was used for the determination of the lipid peroxidation (LPO). One aliquot of homogenate was diluted 1:1 in 0.2 M K-phosphate buffer, pH 7.4, and centrifuged (20 min at 10,000× *g*, 4 °C). Aliquots of the post-mitochondrial supernatant (PMS) were divided into microtubes and kept at −80 °C until the further analyses of their oxidative stress-related biomarkers. From each sample of liver and muscle, three aliquots were used for the analysis of their lipid, sugar and protein contents, and their electron transport system (ETS) activity. All of the biomarker determinations were performed spectrophotometrically, in micro-assays set up in 96-well flat-bottom plates at 25 °C, with the Microplate reader MultiSkan Spectrum (Thermo Fisher Scientific, USA) [22,23].

### 2.4.1. Oxidative Stress-Related Biomarkers

The PMS protein concentration was determined according to the Bradford method [24], using bovine γ-globulin as the standard. The catalase (CAT) activity was determined in PMS by measuring the decomposition of the substrate $H_2O_2$ at 240 nm [25]. The glutathione-S-transferase (GST) activity in the PMS was determined following the conjugation of GSH with 1-chloro-2,4-dinitrobenzene (CDNB) at 340 nm [26]. Total glutathione (tGSH) content was determined in PMS fraction at 412 nm using a recycling reaction of reduced glutathione (GSH) with 5,5′-dithiobis-(2-nitrobenzoic acid) (DTNB) in the presence of a glutathione reductase (GR) excess [27,28]. The tGSH content was calculated as the rate of $TNB^{2-}$ formation with an extinction coefficient of DTNB chromophore formed; $\varepsilon = 14.1 \times 10^3$ $M^{-1}cm^{-1}$ [28,29]. The endogenous lipid peroxidation (LPO) was obtained by measuring thiobarbituric acid-reactive substances (TBARS) at 535 nm [30].

### 2.4.2. Cellular Energy Allocation

The methods described by De Coen and Janssen [31], adjusted for microplates [22], were used to assess the energy available (Ea, as the sum of sugars, lipids, and proteins) and aerobic energy production (Ec, measured as ETS activity), with the final CEA value calculated as CEA = Ea/Ec [32].

The carbohydrates quantification was performed by adding 5% phenol and $H_2SO_4$ to the samples, with glucose as a standard, and the absorbance was read at 492 nm. The total lipid content of each sample was determined by adding chloroform, methanol and ultra-pure water (2:2:1). After centrifugation, $H_2SO_4$ was added to the organic phase of each sample before incubation (15 min, 200 °C). The absorbance was measured at 375 nm, and tripalmitin used as a lipid standard. The total protein content quantification followed Bradford's method [24], with bovine serum albumin as a standard, and the absorbance read at 520 nm. Fractions of the Ea were converted into their energetic equivalent values using the corresponding energy of combustion: 39,500 mJ/g lipid, 17,500 mJ/g glycogen, 24,000 mJ/g protein [33].

The electron transport system (ETS) activity was measured using the INT (Iodonitrotetrazolium) reduction assay, in which ETS is measured as the rate of INT reduction in the presence of the nonionic detergent Triton X-100, with the absorbance read kinetically at 490 nm. The cellular oxygen consumption rate was calculated based on the stoichiometrical relationship, in which—for 2 µmol of formazan formed—1 µmol of oxygen is consumed. The Ec value was obtained by the conversion to energetic values using the specific oxyenthalpic equivalent for an average lipid, protein and carbohydrate mixture of 480 kJ/mol $O_2$ [33].

### 2.5. Statistical Analysis

The data were analysed for normality and homogeneity of variance through D'Agostinho and Pearson's normality test, and the Brown–Forsythe test, respectively. The data were transformed before being treated statistically when necessary. The data were analysed by a two-way ANOVA, with temperature and fish ploidy types as factors. Both procedures were followed by the Šídák multiple comparison post hoc test, in order to identify the differences in the experimental treatments. All of the statistical analyses were performed using GraphPad Prism version 6.0 for Windows (GraphPad Software, La Jolla, CA, USA). The level of significance used was $p < 0.05$ for all of the statistical tests. Full Two-way ANOVA tables are presented in the Supplementary Materials Tables S2–S6.

## 3. Results

### 3.1. O. mykiss Weight Gain

Juvenile fish from both batches presented a significant ($p < 0.0001$) growth during the monitored time, as expected. However, the triploids exhibited significantly ($p < 0.0001$) less fresh weight than the diploid fish, mainly in the last two months of sampling, which was also coincident with the wintertime and colder waters (Figure 2).

### 3.2. Hematology and Immunological Parameters

A decrease in the number of neutrophil cells was observed from late summer to autumn, when the water temperature diminished from 22.0 to 15.4 °C (Table 1). The triploid animals presented a significantly lower number of neutrophils than the diploids ($p = 0.026$), mainly at 22 °C (Table 1). The temperature seasonality significantly ($p = 0.009$) influenced the number of monocytes in the fish, regardless of their ploidy (Table 1). Significant interactions between the temperature and the type of fish ploidy ($p = 0.002$) lead to the triploid fish showing a significant increase in lymphocytes concomitant with a decrease of thrombocytes in the winter (6.6 °C) when compared with their diploid counterparts (Table 1).

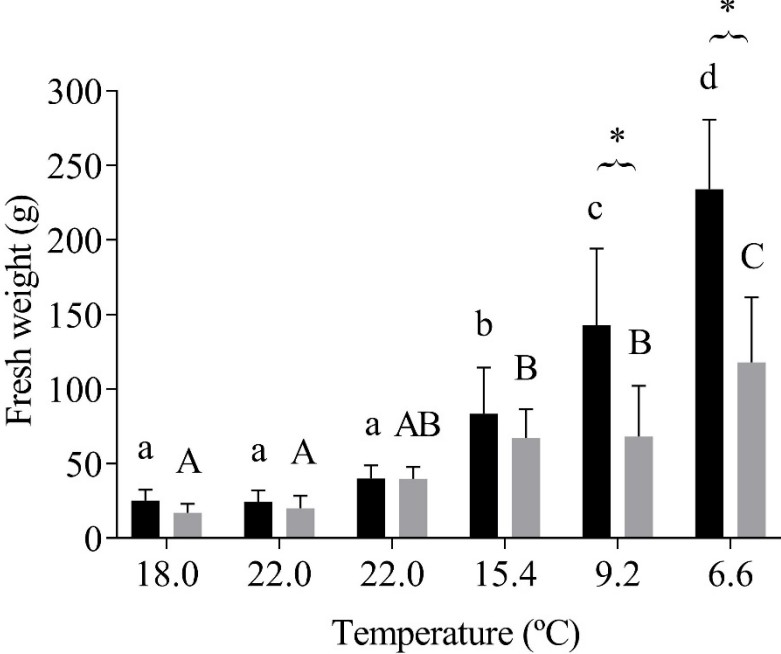

**Figure 2.** Diploid and triploid fish's fresh weight (g) at each sampling temperature. The values are expressed as mean ± SD, *n* = 10. Different letters mean significant differences among water temperature variations to each group; * represents differences between diploid and triploid fish (Sidak post hoc test, *p* ≤ 0.05).

**Table 1.** The relative proportion of diploid (D) and triploid (T) *O. mykiss* peripheral blood leukocytes (thrombocytes, lymphocytes, monocytes, and neutrophils), expressed as a percentage of the total white blood cells (% WBC) at each sampling temperature.

| Biomarker | Ploidy | Temperature (°C) | | | | |
|---|---|---|---|---|---|---|
| | | 22.0 | 22.0 | 15.4 | 9.2 | 6.6 |
| Neutrophils | D | 3.65 ± 2.19 abc | 6.1 ± 6.22 b | 2.1 ± 1.71 c | 3.3 ± 2.72 abc | 4.65 ± 1.63 abc |
| (% WBC) | T | 2.85 ± 2.46 | 1.88 ± 0.52 | 1.81 ± 1.13 | 2.95 ± 1.50 | 4.0 ± 1.71 |
| Monocytes | D | 3.15 ± 2.59 | 0.9 ± 0.57 | 2.2 ± 1.18 | 2.35 ± 2.80 | 2.5 ± 1.29 |
| (% WBC) | T | 3.15 ± 2.37 | 1.35 ± 1.13 | 2.6 ± 1.45 | 2.55 ± 1.92 | 3.3 ± 1.95 |
| Lymphocytes | D | 62.4 ± 8.75 | 61.7 ± 13.12 | 60.9 ± 7.56 | 57.5 ± 7.85 | 59.6 ± 15.6 |
| (% WBC) | T | 52.95 ± 11.68 a | 58.8 ± 7.10 a | 49 ± 9.71 a | 49 ± 10.68 a | 72.8 ± 8.94 b |
| Thrombocytes | D | 30.8 ± 8.11 | 31.3 ± 15.10 | 34.8 ± 6.68 | 36.85 ± 7.70 | 33.25 ± 15.54 |
| (% WBC) | T | 41.05 ± 9.58 a | 36.5 ± 7.98 a | 44.9 ± 10.42 a | 45.5 ± 12.42 a | 19.05 ± 8.09 b |

Values are expressed as means ± SD, n = 10. Different letters mean significant differences due to temperature at the day of sampling; boxes around values mean significant differences between diploid and triploid fish (Sidak's multiple comparisons test).

Significant alterations in antiprotease activity were observed due to temperature (*p* < 0.0001), and among diploid and triploid fish (*p* = 0.007), mainly in winter, at 6.6 °C (Figure 3a). The lysozyme activity was significantly increased after a decrease in water temperature to 6.6 °C in both fish (*p* = 0.0002, Figure 3b). The peroxidase activity was significantly altered by seasonal water temperature changes (*p* < 0.0001), with significant differences between diploid and triploid fish observed at 9.2 °C (Figure 3c).

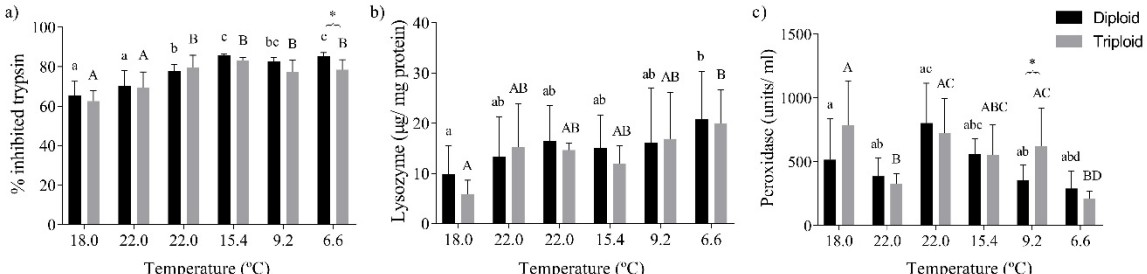

**Figure 3.** Values for the immunological parameters of diploid and triploid O. mykiss, along with seasonal temperature variations: (**a**) antiprotease activity (as % of inhibited trypsin), (**b**) lysozyme activity (µg/mg protein), and (**c**) peroxidase activity (units/mL). The values are expressed as mean ± SD, $n = 10$. Different letters indicate significant differences between temperatures within each group; * represents differences between diploid and triploid fish for each specific group of temperatures, (Sidak post hoc test, $p \leq 0.05$).

### 3.3. Oxidative Stress Status and the Energy Budget of O. mykiss

The CAT activity was significantly changed by the seasonal water temperature ($p < 0.0001$), with lower CAT activity values in diploid and triploid trout collected in the summer months (18.0–22.0 °C) when compared with fish collected from autumn to winter (15.4–6.6 °C; Figure 4a). The fluctuations in the GST activity were also significantly influenced by temperature ($p = 0.0007$, Figure 4b). The tGSH levels were significantly altered by the water temperature ($p < 0.0001$). They were significantly different in diploid and triploid fish ($p < 0.0001$), mainly in the two first sampling times, corresponding to younger animals in warmer waters (18.0–22.0 °C, Figure 4c). Concordantly, the LPO levels were significantly altered by the water temperature ($p < 0.0001$), with the oxidative damage in the liver tissue becoming significantly higher when both fish batches experienced warmer waters (18.0, 22.0, 15.4 °C), contrasting with low oxidative damage at 9.2–6.6 °C (Figure 4d).

The sugar content in the liver tissue varied significantly with both the water temperature ($p < 0.0001$), the fish's type of ploidy ($p = 0.0005$), and their interaction ($p = 0.0001$). The sugar content was significantly lower in fish collected in the autumn and winter (15.4–6.6 °C; Figure 5a). Furthermore, the minor sugar content of triploid fish in their liver tissue was more evident in July, 22.0 °C, when compared with their diploid counterparts (Figure 5a). The livers' lipid content decreased significantly in fish captured from summer until winter (22.0 to 6.6 °C; $p < 0.0001$; Figure 5b). The protein content in the fish livers was significantly changed by the temperature ($p = 0.005$) and fish ploidy ($p < 0.007$). The liver protein content of the diploid fish did not differ significantly over the sampling period. The triploid fish presented higher protein content at 9.2 °C, with a posterior decrease with the lowest water temperature of 6.6 °C. Furthermore, the triploid fish presented higher levels of proteins in the liver than diploids, particularly in the autumn (9.2 °C; Figure 5c). The Ea in the liver was significantly changed by the water temperature ($p < 0.0001$). In summary, for the trends of the individual energy reserves, they were lower in the fish when they were in colder waters (Figure 5d, Table S5). On the contrary, an increase in Ec (estimated as ETS activity) was observed as the water temperature decreased ($p < 0.0001$; Figure 5e). The CEA value was significantly altered with the water temperature ($p < 0.0001$) and its interaction with ploidy ($p = 0.043$). The balance between Ea and Ec culminated in a significant higher CEA for younger diploid fish, although it was lower in the livers of fish exposed to colder waters (Figure 5f).

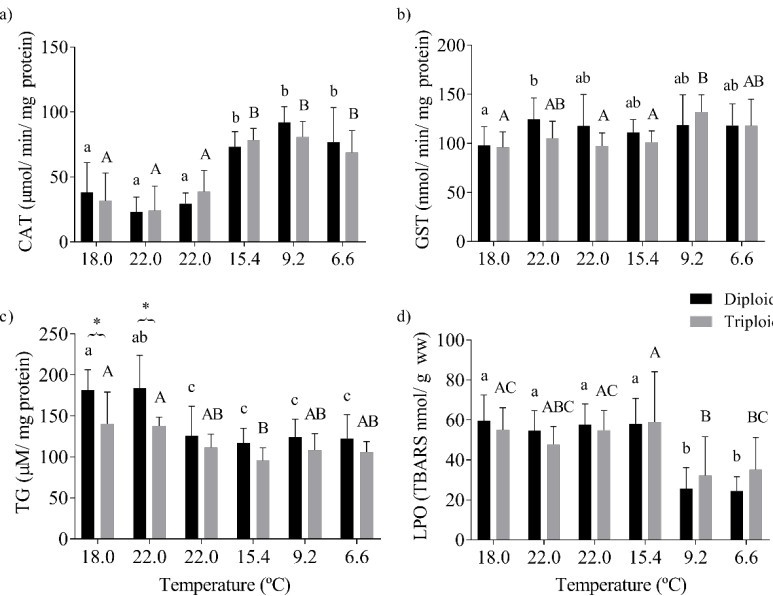

**Figure 4.** Oxidative stress biomarkers in the liver of diploid and triploid O. mykiss, along with seasonal temperature variations: (**a**) catalase activity (CAT, μmol/min/mg protein), (**b**) glutathione S-transferase activity (GST, nmol/min/mg protein), (**c**) total glutathione (TG, μM/mg protein), and (**d**) lipid peroxidation (LPO, nmol TBARS/g wet weight). The values are expressed as mean ± SD, $n = 10$. Different letters indicate significant differences between temperatures within each group; * represents differences between diploid and triploid fish for each specific group of temperatures (Sidak post hoc test, $p \leq 0.05$).

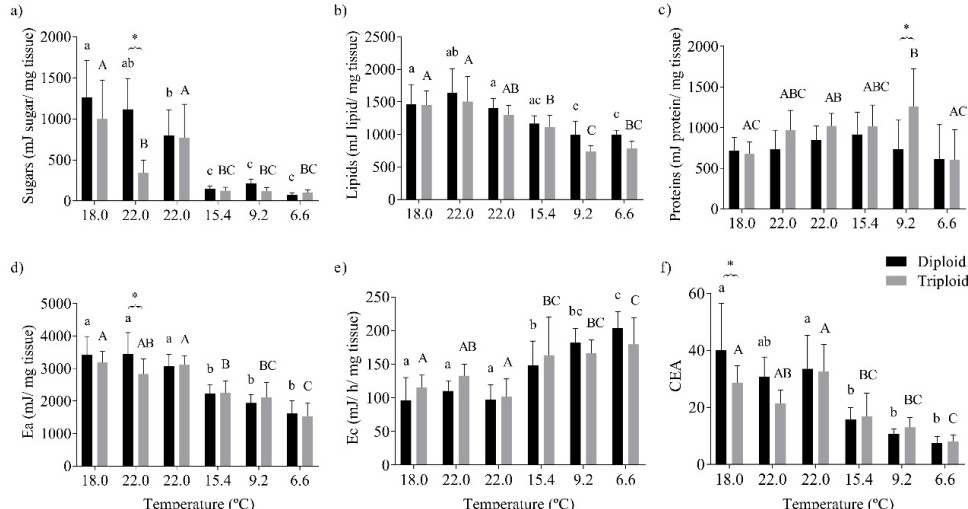

**Figure 5.** Energy budget of the liver tissue of diploid and triploid *O. mykiss* along with seasonal temperature variations: (**a**) sugars (mJ/mg tissue), (**b**) lipids (mJ/mg tissue), (**c**) proteins (mJ/mg tissue), (**d**) energy available (Ea, mJ/mg tissue), (**e**) aerobic energy production (Ec, mJ/h/mg tissue), and (**f**) cellular energy allocation (CEA). The different letters indicate significant differences between the temperatures within each group; * represents differences between diploid and triploid fish for each specific group of temperatures (Sidak post hoc test, $p \leq 0.05$).

The sugar content of the muscle tissue was significantly changed by the seasonal water temperature ($p < 0.0001$) and its interaction with fish ploidy ($p < 0.0001$). The triploid fish showed lower values of sugars than their diploid counterparts at 22.0 and 15.4 °C (Figure 6a). The lipid content increased in the muscle of both fish with the water temperature's ($p < 0.0001$) decrease and its interaction with fish ploidy ($p < 0.0001$). Besides this, larger diploid fish presented the highest lipid content at the end

of the sampling period (6.6 °C; Figure 6b). Seasonal temperature changes ($p = 0.008$) and triploidy ($p < 0.0001$) contributed significantly to alterations in muscle protein content, with triploid fish showing lower levels of proteins (Figure 6c). As a result, the Ea in the muscle tissue was mainly determined by lipids, being significantly altered by the seasonal water temperature ($p < 0.0001$) and its interaction with fish ploidy ($p < 0.002$; Figure 6d). The Ec in the muscle was influenced by the water temperature ($p < 0.0001$), the fish ploidy ($p < 0.006$), and their interaction ($p < 0.028$), with the main differences between both fish groups being observed at 9.2 °C (Figure 6e). The final value of CEA reflected the effects of temperature and their interaction with ploidy, with diploid fish presenting a higher energy budget than triploids at 9.2 °C (Figure 6f).

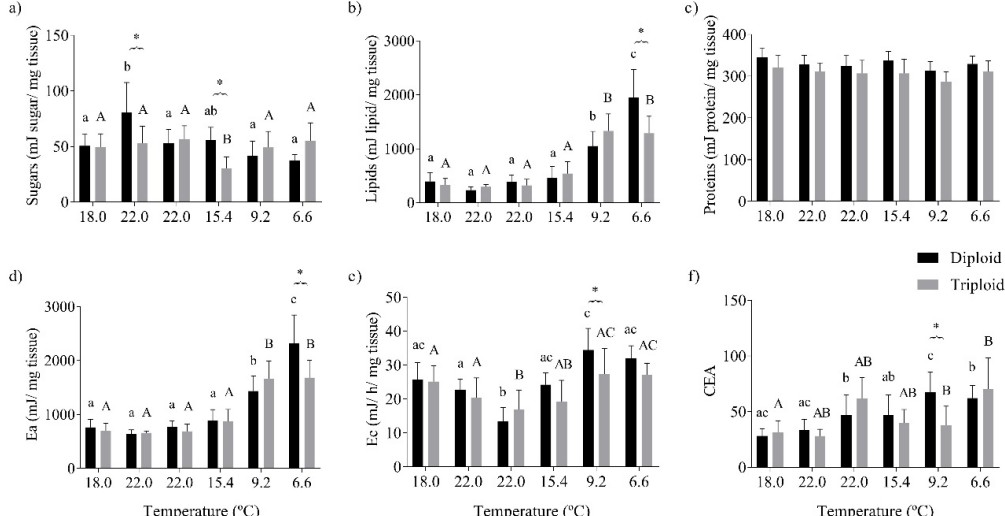

**Figure 6.** Energy budget of the muscle tissue of diploid and triploid O. mykiss, along with seasonal temperature variations: (**a**) sugars (mJ/mg tissue), (**b**) lipids (mJ/mg tissue), (**c**) proteins (mJ/mg tissue), (**d**) energy available (Ea, mJ/mg tissue), (**e**) aerobic energy production (Ec, mJ/h/mg tissue), and (**f**) cellular energy allocation (CEA). Values are expressed as means ± SD, $n = 10$. The different letters indicate significant differences between the temperatures within each group; * represents differences between diploid and triploid fish for each specific group of temperatures (Sidak post hoc test, $p \leq 0.05$).

## 4. Discussion

Nowadays, two main topics of concern are whether climate changes, i.e., surface water temperature increase, are responsible for major losses in aquaculture farms at the northern hemisphere, and whether triploid fish are more advantageous for livestock rearing. Some studies have been developed to answer these questions, but, to the best knowledge of the authors, this is the first study that followed batches of diploid and triploid *Oncorhynchus mykiss* in a real scenario of production in order to give insights into the concerns raised. Thus, the present work depicts the immune status, and biochemical shifts on two independent batches of diploid and triploid *O. mykiss* reared under the same environmental conditions in one Portuguese farm, from June 2016 until January 2017. The seasonal variations include changes in the photoperiod and water quality parameters. Among these, water temperature is known as a preponderant factor shaping the dissolved oxygen and controlling the physiological responses of ectotherm organisms, such as fish [4,34]. Therefore, the data obtained in this study are discussed highlighting water temperature values measured during this period, which varied from 6 to 24 °C, a range of temperature that almost matches the upper limits of thermal tolerance for this species, set as >25 °C [35].

Although seasonal variations contributed to the limited triploid trout weight gain observed after December 2016, future studies with a focus on different growth-related parameters should be developed in farming environments in order to fully clarify the magnitude of the observed differences in the growth of *O. mykiss* individuals. The lower survival and growth of triploid *O. mykiss* under thermal stress has

been demonstrated, particularly in laboratory experiments [6,36]. A similar response was observed due to changes in other environmental conditions, such as rearing in saltwater [37]. Likewise, impaired development has been reported in other induced triploid fish, such as *Acipenser transmontanus* [10] and *Salmo salar* [8].

Fluctuations in water temperature and photoperiod are seasonal events, especially notorious in the temperate zones, known to influence the immunological response of fish [38]. The major changes in the peripheral leucocyte numbers observed in this study can be attributed to these seasonal fluctuations, but the differences in the immune response between the fish ploidies were also noticeable. The seasonal variation in the circulating neutrophils was particularly evident in the summer (22 °C), when the diploids presented more neutrophils than the triploid fish. Regarding lymphocyte counts, diploid trout presented a higher number of these cells compared to triploids until the wintertime. At lower temperatures, the lymphocytes in the triploid fish increased and supplanted the number observed for the diploid fish. Benfey and Biron [39] did not find differences in the number of lymphocytes between diploids and triploids. However, a decrease in the lymphocyte concentrations related to acute handling stress was reported, suggesting that lymphopenia may occur either by a redistribution of lymphocytes, mainly to lymphoid tissues, or by the destruction of lymphocytes following a cytolytic response to higher cortisol levels [39]. Thrombocytes, on the other hand, exhibited the opposite response pattern to the lymphocytes, with its number being higher in triploids until wintertime. In the winter, the number of these cells in the triploids decreased to around 41% as the water temperature reached 6 °C. Therefore, the *O. mykiss* seasonal leukogram changes observed in the present study underline the influence of the water temperature on *O. mykiss* white blood cells. Similar changes in diploid *O. mykiss* have been reported, with enhanced white blood cell counts with increased temperatures (from spring to the summer), and then decreases in autumn, reaching their lowest levels in winter [15,40]. These alterations may be supported by a cell population of juvenile and developing cells observed in rainbow trout in spring and summer months, contrasting with mature cells in winter periods [41]. In fact, water temperature is considered one major factor for leukogram discrepancies. Higher monocyte, granulocyte, and B lymphocyte activation in *O. mykiss* were found between 15 and 17 °C than at lower temperatures (10–12 °C), after inoculation with *Aeromonas salmonicida*. However, the same study reported that a specific antibody response development against infection seemed to be more effective at lower water temperatures [42]. The mechanisms of compensation include higher levels of phagocytosis at elevated temperatures, in order to counterbalance the lymphocytopenia [41]. Moreover, neutrophilic granulocytes are known to be protected by stress, since cortisol inhibits neutrophils apoptosis [43]. Contrary to our findings, neutrophils showed themselves to be the most resistant defense of cells to immunosuppression at low temperatures [44]. Furthermore, the differences between diploid and triploid *Salmo salar* white blood cells in response to vaccine efficacy after infection with *A. salmonicida* were observed, with triploids presenting lower white blood cell numbers [45].

Seasonal fluctuations also altered the humoral innate immune parameters. The antiproteases and lysozyme of rainbow trout achieved higher activities at water temperatures closer to the optimum rearing temperature range (8–15 °C), and were generally higher in diploid fish. Antiprotease activity on *O. mykiss* showed decreased values in the summer months, similarly to the antiprotease activity fluctuations observed in *Gadus morhua* with environmental temperature variations [46]. The water temperature's influence over antiprotease activity was also observed in *Dicentrarchus labrax* [47]. Contrary to the reported pattern, which associates lower temperatures with suppression in the immune parameter, *Sparus aurata* showed decreased cellular and humoral immune parameters at 25 °C [48]. Our results also show that diploid fish produce higher antiprotease mean values than triploid fish. These results are in good agreement with discoveries on *S. salar* [49], suggesting that genetic variations might produce different antiprotease activities. In general, our results indicate that more lysozyme is produced by phagocytic cells at cooler temperatures, improving the fish's immune system, resistance to stress, and infection. The observed increase in plasma lysozymes suggests an activation/triggering of the phagocytic cells, a fact supported by the increase in plasma anti-protease activity. A similar

pattern between white blood cells and lysozyme activity, with increases in the spring/summer months and a subsequent decrease over autumn/winter, has been previously reported for rainbow trout [40]. Peroxidase activity variations do not follow a pattern related to the season; lower values were observed in July 2016 (22 °C) and January 2017 (6 °C). However, higher peroxidase activity in triploid fish compared with diploids was observed at 9 °C.

Regarding the enzymatic antioxidant defenses measured in the liver tissue of rainbow trout, CAT showed a more pronounced decrease in its activity with increasing water temperatures, regardless of fish ploidy. Similarly, negative correlations between CAT activity and water temperatures were observed for juvenile *Salmo marmoratus* [50]. On the other hand, the non-enzymatic antioxidant capacity (tGSH levels) was higher in diploid fish at the two first sampling times in spring/summer, and decreased in both fish in lower water temperatures. Lower hepatic levels of glutathione in triploid juveniles of rainbow trout were also described in response to stress due to transportation [51]. The freshwater *Carassius auratus* is known to tolerate hypoxic warm waters, and previous studies reported an increase in free thiol content under heat shock exposure in this species. The results highlight the role of low-molecular thiols, such as glutathione, in the reduction of protein disulfides, helping to maintain the redox balance and functional state of the proteins [52]. Despite these differences in the antioxidant response of diploid and triploid fish, and contrary to what may be expected, since triploid cells have a higher proportion of membrane lipids susceptible to oxidation [51], no differences were observed due to ploidy in lipid peroxidation. The oxidative damage mirrored the changes in the reactive oxygen species (ROS) production coupled with variations in water temperature [53], with higher LPO levels being observed during the warmer months, and lower levels being measured with the decreasing water temperatures. Fish, as ectotherms, are always subject to changes in water temperature, and have to develop a thermal tolerance to cope with such fluctuations. Dissolved oxygen is known to vary inversely with water temperature, and one of the mechanisms of hypoxia-induced oxidative stress in warmer periods may be related to the reduction of carriers for electron-transport chains due to the limited oxygen available [53,54]. Thus, oxygen consumption implies the rate of ROS generation (review in [34,54]). Therefore, an intensification of respiration can be expected at higher water temperatures, resulting in enhanced ROS production. Consequently, the density and functional properties of the mitochondria occur and are translated into adaptive responses by the antioxidant defenses of the fish when shifts in the water temperatures occur [34], from summer to winter in our case.

The energy demand for detoxification and repair processes led to a decrease in the energy available in the liver tissue, concomitant with an increase in aerobic energy production (estimated as ETS activity), from the warmer to the colder months. The suppression of mitochondrial oxidative phosphorylation and ETS activity is a common outcome of mitochondrial stress, and with fewer electrons available, ROS production increases, and energy deficiency occurs due to the discrepancy among the cellular ATP needed and the mitochondrial ATP production [53]. Further, decreased ETS activity in warmer months may be related to a decrease in the mitochondrial density, diminishing the maintenance costs as proton leaks increase at elevated temperatures. On the contrary, in colder periods, mitochondrial proliferation may support the metabolic compensation, the upkeep of ATP production, and the reduction of the oxygen diffusion distance in the cell [53].

Triploid fish presented lower levels of sugars in their muscle than their diploid counterparts, mainly in July (22 °C) and November (15 °C). These results are in agreement with previous studies that reported that some enzymes related to glycolysis are reduced in triploid *O. mykiss* fry, suggesting an altered metabolic rate in early-stage triploid animals [14]. The lipid content was higher in the last sampling time, corroborating the heavier diploid animals observed. The muscle protein content was likewise influenced by the water temperature, and varied between the two groups of fish, with diploid fish presenting a constant higher value of proteins than triploid fish at all of the sampling times. These results were translated into higher Ea in the muscle of diploid rainbow trout at the end of the monitoring period. The Ec was also higher in diploid animals at lower temperatures (9 °C). The differences in the metabolic enzyme activity of diploid and triploid *Acipenses transmontanus* in

response to warm acclimation also indicated the lower cellular metabolic capacity of triploid individuals under stress conditions [10]. Furthermore, higher metabolic rates were observed for triploid *Salmo salar* and *Salvelinus fontinalis* at lower temperatures (12 °C), along with the inversion of these metabolic tendencies at higher temperatures (18 and 15 °C, respectively) [55].

The capability to tolerate high temperatures of ectotherms is thought to be linked to the maintenance of elevated oxygen delivery to the body, since the temperature alters the aerobic scope as temperatures move to critical values [11]. Our data underline that the weaker performance frequently observed in triploid salmonids due to suboptimal environments, i.e., high water temperatures, is probably due to their lower aerobic metabolic capacity compared to their diploid counterparts [56]. This assumption is based on the reduced surface area to volume ratio (SA:V) in triploid cells and nuclei, which implies a reduction in biochemical and physiological processes, such as oxygen diffusion and signal transport, with consequent limitations on cellular respiration and related processes [45,56,57].

## 5. Conclusions

Fish live in a close balance with the surrounding environment. Naturally, the water temperature—and its inverse relationship with the dissolved oxygen in the water—are preponderant factors influencing the haematological, biochemical, and physiological responses in farmed aquatic animals. Therefore, these parameters are useful and trustworthy indicators for the understanding and prevention of pathological and toxicological impacts on fish. That said, the present work encompasses an extensive evaluation of *O. mykiss* physiological and biochemical alterations under farm conditions during an annual cycle. Thus, it has ecological and economic relevance, since it is a compliment and 'proof of concept' of previous laboratory-based studies, reporting the differences between diploid and triploid *O. mykiss* tracked under the same farm conditions. Therefore, some additional economic and environmental costs due to the lower growth and tolerance of triploid juveniles to changes in environmental conditions might be expected, especially in temperate countries affected by climatic changes. However, the balance between the advantages and disadvantages of cultured triploid fish might be weighted considering several factors, such as the farm's location, broodstock acclimation and epigenetic effects.

**Supplementary Materials:** The following are available online at http://www.mdpi.com/2071-1050/12/21/8785/s1, Table S1: Diploid and triploid *O. mykiss* total lenght (TL, cm) and weight (W, g) measured in farm routine sampling from May 2016 to March 2017, Table S2: Two-way ANOVA results for *O. mykiss* fresh weight with the water temperature at the day of sampling and type of ploidy as factors, Table S3: Two-way ANOVA results for *O. mykiss* peripheral blood leukocytes (thrombocytes, lymphocytes, monocytes, and neutrophils) with water temperature at the day of sampling and type of ploidy as factors, Table S4: Two-way ANOVA results for *O. mykiss* immunological parameters (antiprotease, lysozyme and peroxidase activities) with water temperature and type of ploidy as factors, Table S5: Two-way ANOVA testing for the effects of temperature variation and type of ploidy (D vs T), and their interaction in the biochemical biomarkers in the liver tissue of rainbow trout, Table S6: Two-way ANOVA testing for the effects of temperature variation, type of ploidy and their interaction in the energy budget in the muscle tissue of rainbow trout.

**Author Contributions:** A.C.M.R., C.G., L.E.C.C., B.C., J.C., M.C.-C., A.M.V.M.S., R.J.M.R.: Conceptualisation and Methodology. A.C.M.R., C.J.M.S., S.F.S.P., A.P.L.C., P.S.: Investigation. A.C.M.R., C.G., B.C., R.J.M.R.: Data analysis and Manuscript preparation. L.E.C.C., A.M.V.M.S.: Project administration and Founding acquisition. J.C., M.C.-C.: Fish farm managers. All authors provided intellectual feedback on the manuscript and approved the final version of the manuscript.

**Funding:** Thanks are due to FCT/MCTES for the financial support to CESAM (UIDB/50017/2020+UIDP/50017/2020) and CIIMAR (UID/Multi/04423/2019), through national funds. This work was also supported by Project ALISSA—Healthy and sustainable food for aquaculture fish (reference ALG-01-0247-FEDER-3520), financed by Portugal and the European Union through FEDER, COMPETE 2020 and CRESC Algarve 2020, in the framework of Portugal 2020, and through the COMPETE and Operational Human Potential Programmes, and national funds through Fundação para a Ciência e a Tecnologia (FCT, Portugal). RJMR is funded by national funds (OE), through FCT–Fundação para a Ciência e a Tecnologia, I.P., in the scope of the framework contract foreseen in the numbers 4, 5 and 6 of the article 23, of the Decree-Law 57/2016, of August 29, changed by Law 57/2017, of July 19. FCT also supported BC through grant IF/00197/2015, and APLC through the Ph.D. grant PD/BD/127809/2016.

**Conflicts of Interest:** The authors declare no conflict of interest.

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
