# Peer review of "Seasonal Temperature Fluctuations Differently Affect the Immune and Biochemical Parameters of Diploid and Triploid Oncorhynchus mykiss Cage-Cultured in Temperate Latitudes"

_sustainability, doi:10.3390/su12218785_

Round 1

Reviewer 1 Report

This paper reports on evaluation of seasonal fluctuations on the immune and biochemical status of diploid and triplod rainbow trout, one of the most important aquaculture species, under a realistic aquaculture scenario. The growth, humoral and immune status as well as the oxidative and energy balance were evaluated in different organs reaching the conclusion that diploid fish have a higher tolerance to water temperature changes over the year culminating in a higher growth.

Overall, this is an interesting, new, and original work presenting novel research on this topic and validating previous lab-scale studies. In addition, the presented results might have an economic and ecological impact. However, despite being a well written work, some issues need to be addressed:

  • Review the abstract to include further description of the results obtained. For instance, in the current status it is not perceivable what was really measured nor what changes were observed.
  • Use the italic name the first time the fish species is described, in line 42/43.
  • Include the data for the age and average weight of the fish used.
  • How were the fish sacrificed? Also, animals were collected in the morning or afternoon? Looking at figure 1, there are differences between the same sampling point which could have affected the outcome of the study. Please clarify the temperature at the sampling points.
  • Line 112, review the double concentration of NaHCO3.
  • Line 113, include the concentration/percentage of TCA used.
  • Line 125, include the sigma reference for the lyophilized hen lysozyme.
  • Line 137, clarify which tissue was used for biochemical assays. Also, did the authors equated the analysis of the oxidative-stress markers in the muscle?
  • In the results section, include the p-values for the comparisons made. Also, statistical tables should be removed and the tests included in the main text or moved to supplementary files.
  • Regarding the oxidative stress markers, the first line of defense include other enzymes such as SOD and GPx. For instance, SOD is the first and most powerful detoxification enzyme in the cell. However, it was not measured. In the same way, GPx is another enzyme with high affinity to H2O2 which, in specific cases, respond first than CAT. However, again, it was not measured. Is there any explanation for not conducting such analysis knowing the important role that the set of these enzymes play in the protection against oxidative radicals? Doing such analysis would certainly improve the interpretation of the reported findings.

Author Response

We appreciate the reviewer's comments and made an effort to address them as follows:

  • Review the abstract to include further description of the results obtained. For instance, in the current status it is not perceivable what was really measured nor what changes were observed.

Authors: The abstract was reviewed to include and describe the parameters evaluated. Thanks.

  • Use the italic name the first time the fish species is described, in line 42/43.

Authors: Altered as suggested.

  • Include the data for the age and average weight of the fish used.

Authors: The fish had forth monts at the beginning of the experiment. Information about their average weight and length have been added to the main text.

  • How were the fish sacrificed? Also, animals were collected in the morning or afternoon? Looking at figure 1, there are differences between the same sampling point which could have affected the outcome of the study. Please clarify the temperature at the sampling points.

Authors: Fish were sacrificed by an overdose of anaesthetic. The animals were collected in the morning, to accompany the aquaculture routine, and the values of temperature indicated in figures were measured at the time of sampling. This information has now been added to the main text of the MM section. Please see Lines 85-93.

  • Line 112, review the double concentration of NaHCO3.
  • Line 113, include the concentration/percentage of TCA used.
  • Line 125, include the sigma reference for the lyophilized hen lysozyme.
  • Line 137, clarify which tissue was used for biochemical assays. Also, did the authors equated the analysis of the oxidative-stress markers in the muscle?

Authors: Information has been clarified. Thanks.

  • In the results section, include the p-values for the comparisons made. Also, statistical tables should be removed and the tests included in the main text or moved to supplementary files.

Authors: We changed the results section accordingly, p-values were now added to the main text and ANOVA tables are shown in a supplementary file.

  • Regarding the oxidative stress markers, the first line of defense include other enzymes such as SOD and GPx. For instance, SOD is the first and most powerful detoxification enzyme in the cell. However, it was not measured. In the same way, GPx is another enzyme with high affinity to H2O2 which, in specific cases, respond first than CAT. However, again, it was not measured. Is there any explanation for not conducting such analysis knowing the important role that the set of these enzymes play in the protection against oxidative radicals? Doing such analysis would certainly improve the interpretation of the reported findings.

Authors:  We partially agree with the reviewer, but due to the sample restrictions and in order to have an overview of the antioxidant response capacity of the fish liver, we decided to measure CAT activity. The objective is to understand what happens in terms of antioxidant capacity in relation to peroxide, since, in the presence of essential metals (such as Fe, Zn), the peroxide is converted into the hydroxyl radical responsible for the oxidative damage in macromolecules. In addition, CAT is known an the enzyme with highest turnover and affinity for peroxide. In turn, GPx's affinity for peroxide depends on the isoform we are measuring.

Reviewer 2 Report

The authors of this study attempt to give a broader insight into physiological and metabolic differences between diploid and triploid Oncorhynchus mykiss under intensive culture conditions and simulated, to actual, seasonal fluctuations. They did so by evaluating the effects of seasonal variation on growth, humoral and immune status in blood and plasma (studying peroxidase,  lysozyme and antiproteases activities), oxidative stress balance in liver (by evaluating antioxidant defence, my measuring catalase and glutathione-S-transferase activity, total glutathione, and lipid peroxidation), energy budget in liver and muscle of diploid and triploid (in reference to different lipid and protein content) of O. mykiss. The study is scientifically and technically sound, and experimentally well designed, and the performed experiments are quite suitable to the set aims. The manuscript is concisely written, and the conclusions drawn are supported by the data (figure and tables), and advance further the knowledge in this area, as represented by adequate referencing of past studies.

Author Response

We appreciate the reviewer appreciation of our manuscript and work.

Reviewer 3 Report

This manuscript touches on the important topic of sustainability of fish farming in sub-optimal temperatures. This is an important topic affected by climate change and the thermal optimum of the species as well as variation in ploidy. The scientific approaches used here to understand changes in metabolism, immunity and oxidative stress are relevant and interesting. However my major criticism of the work is in regards to the growth data in Figure 2, used to infer various interpretations with regards to performance of the fish. The data relate to 10 fish per sampling point out of what appears to be 2 cages of fish. This is not a suitable representation of the overall fish population and may also be pseudoreplicated if only 1 cage was used per fish type. While the sampling regime is ok to carry the advanced analyses, the overall cage growth data should be provided to ascertain that fig 2 is representative. I also would like to see greater emphasis on the culture parameters such as feed intake changes that could explain the changes in body reserves (muscle and liver). We note that some of the co-authors are the farm managers so do believe these data can be provided as part of major revisions. Some comments are provided in the pdf. 

Author Response

Authors: We appreciated the criticism of the reviewer and made an effort to addressed all the concerns.

Regarding the data for fish growth, we now include a supplementary file with a table with weight and length of fish from April 2016 to March 2017, with the routine farm measurement of 10 individuals per sampling time.

These cages, part of the farm facilities, are floating cages located in the fourth bigger dam of Portugal. Feed of animals followed the farm routine, with an average density of 12-15 kg/ m3, fish were fed twice a day with 0.7 to 2.3 % alpis extruded fed, as indicated in the MM section. Therefore, a relative feeding intake is evaluate by the farmers based on these metrics.

We understand the concern of the reviewer, but, one of the objectives of this work was to follow the same batch of fish throughout the year. As the batch of diploids and triploids were divided into two cages, respectively, we sampled the fish in these cages. When the main focus of the work is to evaluate the physiological changes of farmed animals in a real scenario of production, sometimes this scheme of sampling is followed. Please see another work with similar sampling procedures and cage number: https://www.nature.com/articles/s41598-019-45657-3

The experimental design was previously submitted for approval by the competent national authorities, DGAV. The number of animals used was calculated with a statistical basis in order to guarantee a high possibility of detecting the effect of the studied variables. The choice of n = 10 represents a compromise between the minimum number of organisms necessary to have statistical strength in the analysis of the results, considering the sampling points and density of animals in the production, as well as the guidelines for animal sampling considering the principles of the 3Rs (Replacement, Reduction and Refinement).

L 43: check value, I am sure its closer to a million tonnes based on value?

Authors: The value cited is available at: http://www.fao.org/fishery/culturedspecies/Oncorhynchus_mykiss/en

Round 2

Reviewer 3 Report

Thanks for the revision and providing the farm data. Also note the FAO site is clearly stating 814,000 tonnes, not 814 tonnes as you have stated on L44. I still believe there is a limitation of your growth data, in that it is pseudoreplicated as only 1 cage was used per fish type/ploidy. The paper you noted did not investigate growth. I would like to see additional discussion with regards to the growth results limitation (potential cage effects as a factor not accounted for here) as based on comparing only 1 cage against 1 cage, let alone 10 fish per cage. Some of the growth differences could come from cage specific issues beyond the scope of this paper. 

Author Response

We would like to thank the reviewer for this second opinion on our manuscript.

Thanks for the revision and providing the farm data. Also note the FAO site is clearly stating 814,000 tonnes, not 814 tonnes as you have stated on L44.

Authors: The value has now been corrected. Thanks.

I still believe there is a limitation of your growth data, in that it is pseudoreplicated as only 1 cage was used per fish type/ploidy. The paper you noted did not investigate growth. I would like to see additional discussion with regards to the growth results limitation (potential cage effects as a factor not accounted for here) as based on comparing only 1 cage against 1 cage, let alone 10 fish per cage. Some of the growth differences could come from cage specific issues beyond the scope of this paper. 

Authors: We understand the point of view of the reviewer and altered the emphasis on growth in the sections of the manuscript (please see Subtitle 3.1 that now reads O. mykiss weight gain). We also made a safeguard in the discussion, for the need for future studies focusing on growth (please see Lines 327-330).

Nevertheless, allow us to clarify that as the sampling period comprised almost one year-round, and these fish were from two regular farmed batches, calibrations for fish growth/density were performed routinely in each cage with 16 meters of diameter, with an average density of 12-15 kg/ m3. Sampling days for our work were at distinct and spaced days to avoid some well-known effect of handling stress.